# Serum Levels of miR-34a-5p, miR-30b-5p, and miR-140-5p Are Associated with Disease Activity and Brain Atrophy in Early Multiple Sclerosis

**DOI:** 10.3390/ijms26178597

**Published:** 2025-09-04

**Authors:** Riccardo Orlandi, Leopoldo Torresan, Francesca Gobbin, Elisa Orlandi, Macarena Gomez Lira, Alberto Gajofatto

**Affiliations:** Department of Neurological Sciences, Biomedicine and Movement Sciences, University of Verona, Piazzale Ludovico Antonio Scuro 9, 37134 Verona, Italy; riccardo.orlandi@univr.it (R.O.); leopoldo.torresan@studenti.univr.it (L.T.); francesca.gobbin@univr.it (F.G.); elisa.orlandi@univr.it (E.O.); macarena.gomezlira@univr.it (M.G.L.)

**Keywords:** multiple sclerosis, biomarkers, microRNA, miRNA, magnetic resonance imaging (MRI)

## Abstract

In recent years, research has focused on biomarkers as key tools to predict clinical outcomes and guide therapeutic decisions in Multiple Sclerosis (MS). MicroRNAs (miRs)—small non-coding RNA molecules that regulate gene expression at the post-transcriptional level—have emerged as promising biomarkers in MS due to their accessibility in biological fluids. This study investigates the role of specific serum miRs mainly involved in immune response regulation as potential prognostic biomarkers in MS, focusing on young patients with recent diagnosis. The study had a prospective design, involving a cohort of patients followed in the Hub and Spoke MS network of Verona province. Fifty-one patients (33F) aged 18–40 years with recent MS diagnosis (≤2 years; 45 relapsing-remitting, 6 primary progressive) were consecutively enrolled. At baseline, serum samples were collected for miR analysis alongside clinical-demographic and MRI data, including T2 lesion volume, normalized brain volume (NBV), gray matter volume, white matter volume (WMV) calculated at baseline and annual percentage brain volume change (PBVC) and occurrence of new T2 or gadolinium-enhancing (Gd+) lesions on follow-up scans. Candidate miRs were chosen based on their potential biological role in MS pathogenesis reported in the literature. miRs assays were done using real-time PCR and expressed as a ratio relative to a normalizer (i.e., miR-425-5p). Levels of miR-34a-5p were significantly higher in patients with Gd+ lesions (*p* < 0.001) and correlated to lower NBV (rho = −0.454, *p* = 0.001) and WMV (rho = −0.494, *p* < 0.001). Conversely, miR-140-5p exhibited a protective effect against occurrence of new T2 or Gd+ lesions over time (HR 0.43; IC 95% 0.19–0.99; *p* = 0.048). Additionally, miR-30b-5p correlated directly with PBVC (adjusted rho = −0.646; *p* < 0.001). These findings support the potential of serum miR-34a-5p, miR-140-5p, and miR-30b-5p as markers of disease activity and progression in patients with recently diagnosed MS.

## 1. Introduction

Multiple sclerosis (MS) is a complex immune-mediated disease of the central nervous system (CNS): its clinical course reflects the interplay between acute and chronic inflammation processes, neuroaxonal injury and repair mechanisms that vary among individuals and within the same individual over time, accounting for the clinical heterogeneity of this condition [1].

Consequently, there is significant interest in identifying biomarkers that can provide an objective and quantifiable measure of the pathological processes active in individual patients, with the goal of predicting the clinical evolution of the disease and optimizing treatment [2]. Current clinical practice is provided with relatively limited tools to assess prognosis and response to therapy in MS at the individual level, including: (1) socio-demographic factors (e.g., male gender, non-Caucasian ethnicity, obesity, and cigarette-smoking, which are associated with a poor clinical course); (2) presentation features (e.g., spinal cord and/or brainstem syndromes, motor symptoms, multifocal manifestations, frequent relapses, poor clinical recovery, and progressive course, which are associated with an increased risk of neurological disability); (3) brain and spinal cord magnetic resonance imaging (MRI) measures (e.g., high number of demyelinating lesions and presence of contrast-enhancement that are predictive of subsequent relapses; lesion location in the brainstem, spinal cord and cerebral cortex as well as brain and spinal cord atrophy, which are associated with disability progression; (4) laboratory findings such as presence of IgG oligoclonal bands restricted to the cerebrospinal fluid; and (5) neurophysiological measures such as visual, somato-sensory and motor evoked potential abnormalities, which are associated with a possible increased risk of long-term disability [3]. However, most of these variables are non-scalable and retrospective risk factors, as for demographic features and clinical presentation, which are not particularly helpful as predictive factors at the individual level. Other available measures are not linked to specific biological processes or not frequently repeatable over time due to acceptability, time and cost constraints, as in the case of MRI, conventional cerebrospinal fluid analysis, and evoked potentials.

In this context, microRNAs (miRNAs), which are small non-coding RNAs, could represent an ideal predictive biomarker in people living with MS (pwMS) since they are involved in regulating crucial biological processes in MS pathogenesis by acting as post-transcriptional modulators of gene expression. Moreover, they can be detected with high sensitivity in easily accessible body fluids such as blood, saliva, and urine, where they maintain high stability even under extreme temperature and pH conditions [4,5]. Therefore, circulating miRNAs detection appears more targeted to specific molecular pathways, more accurately quantifiable, and more repeatable compared to currently available tools for assessing prognosis and monitoring disease evolution in MS.

For instance, miR-325 expression is implicated in the regulation of cell invasion and proliferation, which represent key processes in the inflammatory cascade driving the pathogenetic mechanisms of MS [6]. Several studies have investigated serum concentration of circulating miRNA, allowing the identification of different alteration profiles in MS patients. However, few studies have prospectively examined the role of miRNAs expression in predicting the clinical and radiological evolution of the disease in people with recent diagnosis of MS. It has been shown that blood expression of miR-30b-5p, miR-140-5p, miR-122-5p, miR-196b-5p, and miR-432-5p may help to differentiate between MS patients and healthy individuals or patients with other neurological conditions [7,8,9,10,11,12]. Other studies have indicated the potential ability of miR-155, miR-128-3p, and miR-34a-5p serum concentration in separating MS patient groups with different clinical courses [13,14,15,16,17]. Evidence on the utility of miRNA expression to assess treatment efficacy is scarce and controversial. Karimi et al. showed similar levels of miR-326-3p in blood lymphocytes of MS patients with and without good clinical response to interferon-beta therapy, in contrast to previous reports showing miR-326-3p down-regulation in peripheral blood mononuclear cells of interferon-beta responders [18].

Given this background, the aim of the present study was to prospectively explore the potential relationship between consolidated measures of disease activity and progression in patients with early-stage MS and serum levels of the above-mentioned candidate miRNAs, which are involved in immune system regulation mechanisms potentially relevant to MS pathogenesis [7,8,9,10,11,12,13,14,15,16,17,18].

## 2. Results

Fifty-one patients were consecutively enrolled as outlined in the flowchart in Figure 1 and according to the inclusion criteria reported in the Methods section:5 patients did not have a baseline brain MRI suitable for volumetric and lesion load assessment and were therefore excluded from MRI measures analysis.1 patient was lost to follow-up and was thus excluded from the prospective data analysisin a subgroup of 28 patients with MRI at 24 months in accordance with the study protocol, it was possible to assess the annual percentage brain volume change (PBVC).

Main demographic, clinical, and MRI variables for the entire cohort are reported in Table 1.

This cohort consists of patients with a mean age of 33 years, predominantly female (64.7%), and with a low degree of disability (median expanded disability status scale (EDSS) score of 1.5, range 1–4). At baseline, approximately one-fifth of the patients had gadolinium-enhancing lesions on brain MRI.

These patients were followed with clinical and MRI monitoring for a period ranging from 17 to 75 months (median 45 months). During this follow-up, 36% of the patients experienced clinical relapses, and 48% showed MRI disease activity. Additionally, 30% of the patients had a confirmed increase in their EDSS score, and 92% initiated disease-modifying therapy.

The subgroup of patients included in the PBVC analysis did not differ significantly from patients of the entire prospective cohort with respect to the collected variables (Table 1).

### 2.1. Expression of miRNA-34a-5p, Gadolinium-Enhancing Lesions, and Brain Volume Measures

Serum levels of miR-34a-5p were significantly higher in patients with at least one active (gadolinium-enhancing) lesion on brain MRI performed at baseline (median miR-34a-5p: 0.164 vs. 0; *p*-value < 0.001, see Figure 2).

A significant inverse correlation was also observed between miR-34a-5p expression and both global brain volume (normalize brain volume (NBV), rho = −0.454, *p* = 0.001) and white matter volume (WMV, rho = −0.494, *p* < 0.001; see Figure 3).

This finding was also confirmed in the partial correlation analysis adjusted for sex, age, EDSS, presence of gadolinium-enhancing lesions on baseline brain MRI, and disease duration (with rho = −0.443, *p* = 0.004 for NBV; rho = −0.481, *p* = 0.002 for WMV, respectively).

Moreover, a significant direct correlation emerged between miR-34a-5p levels and serum light chain neurofilaments (with rho = 0.289; *p* = 0.042).

This data, however, is not confirmed in the partial correlation analysis (adjusted for the same variables as shown before). The expression of no other miRNA was found to be correlated with neurofilament levels in the univariate analysis.

### 2.2. Expression of miR-140-5p and MRI Disease Activity

Serum levels of miR-140-5p were significantly higher in patients who did not develop MRI disease activity during the follow-up period (median miR-140-5p: 0.223 vs. 0; *p* = 0.034; see Figure 4).

To evaluate the relationship between miR-140-5p expression and the occurrence of disease activity on follow-up MRIs, a survival analysis was performed using the Kaplan–Meier method (see Figure 5).

The risk of developing new T2 or gadolinium-enhancing lesions during follow-up was significantly lower in patients with higher serum levels of miR-140-5p (HR 0.43; 95% CI 0.19–0.99; *p* = 0.048).

Patients with higher expression of this miRNA exhibited a significantly longer time to event occurrence compared to those with lower expression levels.

A multivariate analysis was then conducted using the Cox regression model, accounting for age, sex, presence of gadolinium-enhancing lesions on baseline MRI, disease duration, and concomitant treatment with disease-modifying therapy.

As shown in Figure 6, this analysis did not confirm the findings of the univariate analysis, although it suggested a trend toward a protective effect of miR-140-5p against the development of new T2/Gd+ lesions. However, this did not reach statistical significance (HR 0.45, *p* = 0.092).

### 2.3. Expression of miR-30b-5p and PBVC

The expression of miR-30b-5p showed a significant direct correlation with annual brain volume loss as measured by PBVC (rho 0.629; *p* < 0.001, see Figure 7).

This data was also confirmed in the partial correlation analysis adjusted for age, sex, EDSS, disease duration, and the presence of gadolinium-enhancing lesions at baseline (rho—0.646 and *p* < 0.001). Therefore, higher serum levels of miR-30b-5p at baseline are associated with greater global brain volume loss at 24 months.

No statistically significant associations were found between miR-34a-5p, miR-140-5p, miR-30b-5p expression and the other variables collected in the study. No statistically significant associations were also found between the expression of remaining miRNAs included in the panel and the examined clinical and MRI variables.

Finally, no statistically significant differences were found in the examined miRNA levels between MS cases and controls.

## 3. Discussion

In this study, we explored the possible relationship between serum levels of candidate miRNAs selected for their biological roles and variables of disease activity and progression in a prospective cohort of patients with a recent diagnosis of MS.

### 3.1. MiR-34a-5p

We found a significantly higher expression of miR-34a-5p in patients who presented with at least one gadolinium-enhancing lesion on baseline MRI. Additionally, the levels of this miRNA showed a direct correlation with the degree of global brain atrophy and white matter loss. Furthermore, this miRNA was the only one found to be directly correlated with neurofilament light chain serum levels at baseline. These results suggest a potential role of miR-34a-5p as a marker of inflammatory activity and neuroaxonal loss. In line with this hypothesis, previous evidence indicates that patients with MS show significantly elevated levels of miR-34a-5p during relapses, which gradually decrease and normalize during disease remission. It has been reported that miR-34a-5p can alter T-cell activity by promoting a pro-inflammatory response through the dual inhibition of Treg cells and induction of a Th17 phenotype [19]. Furthermore, miR-34a-5p is overexpressed in active lesions of MS patients, where, in synergy with miR-155, it contributes to reduced expression of CD47. This protein, present on the surface of oligodendrocytes and astrocytes, serves as an inhibitory signal for the phagocytic action of macrophages. The absence of this signal, driven by the overexpression of miR-34a-5p, would promote demyelination and the development of axonal damage, a common feature of active lesions, thus potentially triggering neurodegenerative processes [13]. A recent study aimed at identifying potential miRNAs predictive of transition to SPMS found that miR-34a-5p is overexpressed in RRMS patients who progress in the subsequent 10 years, making it a potentially valuable marker of a more severe disease course [14]. Furthermore, this finding further confirms the close relationship between inflammation and neurodegeneration in MS.

### 3.2. MiR-140-5p

Our study revealed that MS patients who developed new T2 or gadolinium-enhancing lesions on follow-up brain MRI had lower expression of miR-140-5p. This suggests a potential utility of this miRNA as a proxy of reduced inflammatory MRI activity in MS. Previous studies have highlighted that reduced expression of miR-140-5p is a distinguishing feature of MS patients compared to healthy individuals, suggesting its potential as a diagnostic marker [10,14]. This miRNA regulates the inflammatory response by modulating T-cell differentiation toward the Th1 phenotype via inhibition of the STAT1 pathway. Lack of miR-140-5p expression would increase Th1 cell shift and inflammatory activity. In vitro studies have shown that T-cells from MS patients exhibit significantly reduced conversion to the Th1 phenotype when transfected with miR-140-5p, thus opening the door to potential therapeutic applications [20]. It is worth noting that in our study, we observed an association between miR-140-5p and MRI disease activity but not with clinical relapses. This discrepancy may reflect the greater sensitivity of MRI, as evidenced by the higher number of patients with new T2/Gd-enhancing lesions during follow-up compared to those with clinically defined relapses.

### 3.3. MiR-30b-5p

In patients with overexpression of miR-30b-5p, we observed a direct correlation with annual brain volume loss (PBVC/year). In a study exploring the potential intersection between MS and the most common neurodegenerative diseases, i.e., Alzheimer’s disease, Parkinson’s disease, and Amyotrophic Lateral Sclerosis, miR-30b-5p was the only miRNA overexpressed across the three disease groups [7]. In light of these data, it is possible to hypothesize that this miRNA, when overexpressed in MS patients, may serve as a marker of neuroaxonal damage [21]. A recent study evaluating the molecular expression profile changes in MS patients after starting cladribine therapy showed a reduction of this miRNA at 3 and 12 months after treatment initiation. The authors also observed, through proteomics techniques, a modification of the expression of some of the target genes of this miRNA. Therefore, it is possible to assume that miR-30b-5p could not only be a predictive marker of treatment response but also involved in disease regulation mechanisms [22].

### 3.4. MiR-128-3p

In previously reported preliminary data of our research group, miR-128-3p was more highly expressed in patients with a progressive phenotype compared to those with RRMS. Additionally, an inverse correlation was observed between miRNA levels and the annual relapse rate [16]. However, these results were not replicated in the present study. The lack of confirmation of the correlation with the progressive phenotype may be attributed to the limited number of patients with progressive forms in this study. Regarding the second point, it should be noted that while several studies agree on the association between miR-128-3p and progressive forms of MS, the relationship with relapses remains controversial. Other studies, for instance, have reported an increased number of relapses in patients with overexpression of miR-128-3p [15]. Based on these data, it is possible to state that miR-128-3p primarily acts in mechanisms related to disease progression, with a variable or unclear role regarding relapse risk, justifying the conflicting evidence observed across different studies.

### 3.5. Limitations

Our study has several limitations:Selection of the miRNA panel analyzed based on a candidate biomarker approach by biological roles or previous evidence rather than using a deep sequencing method, which however would have required a much larger sample size than in the present study.Small sample size, especially regarding the longitudinal evaluation of PBVC, may have negatively impacted statistical power.Heterogeneity in disease duration as measured by follow-up time since symptoms onset may have compromised comparability of study participants, though this was mitigated by selecting patients with a young age range, recent diagnosis, low disability level, and no exposure to disease-modifying therapies at enrollment.The study would have benefited from the inclusion of cytokine profiling, which may have shed more light on the immune mechanisms driving the disease process and would have allowed to assess whether changes in miRNA expression levels reflected alterations in cytokine expression levels. Although this analysis was not performed due to research protocol restrictions, it remains a future perspective of the research group.

## 4. Materials and Methods

The inclusion criteria for the study were:-Age between 18 and 40 years.-MS diagnosis according to the 2017 McDonald criteria within the previous two years.-No exposure to any DMT approved for MS (i.e., alemtuzumab, azathioprine, cladribine, cyclophosphamide, dimethyl fumarate, fingolimod, glatiramer acetate, interferon beta 1-a, interferon beta 1-b, mitoxantrone, natalizumab, ocrelizumab, ofatumumab, ozanimod, ponesimod, rituximab, siponimod, and teriflunomide) before and at the time of serum sampling (i.e., study enrolment visit).-No exposure to steroid therapy within 30 days before enrollment.-A brain MRI scan performed within six months before or one month after enrollment.-A serum sample collected within two months of inclusion for miRNA and neurofilament light chain analysis.

Patients were enrolled between 2019 and 2022 and they underwent clinical evaluation at the enrollment and every six months, with periodic brain MRI scans every 12 months. Clinical and MRI data were analyzed in relation to the expression of a miRNA panel that included a group of nine miRNAs selected on the basis on their suggested biological role in the literature (see “Introduction” and “miRNA Analysis” sections) [7,8,9,10,11,12,13,14,15,16,17,18].

Patients enrolled in the study were part of a broader project exploring a wide range of clinical, biological and psychosocial characteristics of young adults with a recent diagnosis of MS conducted at Verona University, i.e., BPS-ARMS project [23].

### 4.1. Clinical Assessment

Patients underwent the following clinical assessments:-At inclusion demographic variables, disease onset date and clinical manifestation, clinical course (relapsing-remitting or progressive), and EDSS score were collected.-During follow-up, patients were evaluated every six months for new clinical relapses or significant EDSS changes.-The annualized relapse rate (ARR) was calculated as the ratio of total relapses to observation years for each individual.-At each visit, data on the initiation, discontinuation, or modification of DMT were collected.

Clinical relapses were defined as the appearance of new neurological signs and/or symptoms consistent with a demyelinating event lasting at least 24 h in the absence of fever or infection. Significant EDSS changes were defined as an increase of at least 1.5 points if baseline EDSS was 0, 1 point if between 1 and 5, and 0.5 points if above 5, confirmed at a subsequent assessment at least six months later.

### 4.2. MRI Protocol

For all patients brain (and cervical, when available) MRI scans performed within the inclusion timeframe were acquired according to a standardized protocol:-Brain MRI: volumetric T1 pre- and post-contrast sequences, 3D FLAIR, and 3D double inversion recovery (DIR).-Cervical cord MRI: sagittal and axial scans with volumetric T1 pre- and post-contrast sequences, T2-weighted, and short tau inversion recovery (STIR) sequences.

Acquired images were analyzed using dedicated software for brain volume processing. Normalized brain volume (NBV), white matter volume (WMV), and gray matter volume (GMV) were measured using SIENAX, an automated tool within the FSL package (https://web.mit.edu/fsl_v5.0.10/fsl/doc/wiki/SIENA(2f)UserGuide.html, accessed on 31 August 2025). T2 lesion volume (T2LV) was measured using the semi-automated ITK-SNAP software version 4.0 (https://www.itksnap.org/pmwiki/pmwiki.php, accessed on 31 August 2025), which segments lesions based on 3D FLAIR sequences. Images were reviewed and if needed manually corrected by an investigator from the Verona MS center (RO).

Patients enrolled in the study underwent annual MRI scans to monitor disease activity and calculate the percentage of brain volume change (PBVC). Disease activity was defined by the appearance of new T2-hyperintense lesions compared to baseline and/or contrast-enhancing T1 lesions in the brain or spinal cord. T2 lesion enlargement alone was not considered due to its low inter-observer agreement in clinical practice.

PBVC was calculated using SIENA (https://web.mit.edu/fsl_v5.0.10/fsl/doc/wiki/SIENA(2f)UserGuide.html, accessed on 31 August 2025), another FSL software tool that provides a reproducible percentage change analysis of brain volume between two scans. Baseline MRI scans were compared with follow-up scans acquired using the same MRI equipment at an interval of 24 ± 12 months, and PBVC values were annualized for comparability.

### 4.3. MiRNA Analysis

The study analyzed the following miRNAs: miR-128-3p, miR-30b-5p, miR-34a-5p, miR-122-5p, miR-196b-5p, miR-326-3p, miR-432-5p, miR-155-5p, and miR-140-5p based on previously published data [7,8,9,10,11,12,13,14,15,16,17,18].

Serum samples stored in the same laboratory and collected from 17 individuals (9 females, 8 males, mean age 51 ± 18 years) with non-MS related conditions were used as controls (5 dementia, 4 peripheral neuropathy, 2 autoimmune encephalitis, 1 lymphoma, 2 cerebrovascular disease, and 3 movement disorder cases).

miRNAs were extracted from 200 µL serum samples stored at −80 °C. Frozen samples were thawed on ice and miRNAs were extracted using the Norgen total RNA purification kit (Norgen Biotek, Thorold, ON, Canada) and subjected to reverse transcription using the TaqMan™ Advanced miRNA cDNA Synthesis Kit (Thermo Fisher Scientific Inc., Waltham, MA, USA). Individual miRNAs were identified and quantified via real-time PCR.

Reverse transcription reactions of RNAs were performed using the TaqMan miRNA Reverse Transcription Kit and miRNA-specific stem-loop primers (Applied BioSystems, Waltham, MA, USA) in a scaled down (5 μL) RT reaction, using a fixed volume (1.67 μL) of total RNA from each sample. Real-time PCR was performed on the diluted (1:2) cDNA template with assay-specific primers and probes, using TaqMan technology (Applied Biosystems, Waltham, MA, USA) in final volumes of 10 μL. Each amplification reaction was performed in triplicate wells. Data were normalized using the 2^−ΔCt^ method for relative quantification: ratio = relative quantity (RQ) miR target/RQ miR reference, where RQ = 2 − (Ct sample − Ct control) and Ct control = mean Ct of all samples for a specific miR [24]. Real-time PCR was performed on the CFX Connect Real-Time System (Bio-Rad, Hercules, CA, USA).

Among five tested potential normalizers (i.e., miR-27a-3p, miR-15b-5p, miR-128-3p, miR-20a-5p, and miR-425-5p), miR-425-5p was the most stable, as determined using BestKeeper software (version 1), and it was chosen as reference (Figure 8). BestKeeper is based on repeated pairwise correlation analysis.

In the Results section miRNA levels are expressed as the ratio of candidate miRNA concentration to the normalizer resulting from the above-mentioned formula.

### 4.4. Serum Neurofilament Light Chain Analysis

Serum samples were clotted for 30 min at room temperature and then centrifuged, aliquoted at room temperature, and stored at −80 °C. The temperature of the freezers was continuously monitored and samples were thawed only prior to analysis. Measurement of NfL concentration was performed in both study phases using SIMOA Nf-light^®^ kit in SR-X immunoassay analyzer, Simoa™ (Quanterix Corp, Boston, MA, USA), which runs ultrasensitive paramagnetic bead-based enzyme-linked immunosorbent assays. Briefly, frozen samples and calibrator were equilibrated to room temperature and diluted with specific sample diluent. Calibrators, samples, detector, and beads were dispensed in each well and plates were incubated at 30 °C with shaking 800 rpm for 30 min. After washing steps, 100 μL SBG was added to each well and plates were incubated at 30 °C with shaking 800 rpm for 10 min. After washing steps, beads were resuspended twice at 1000 rpm for 1 min. A final washing step was performed and plates were dried for 10 min before being transferred to the SR-X for reading.

### 4.5. Statistical Analysis

Normally distributed continuous variables were reported as mean ± standard deviation, while those with non-normal distribution were described as median and range or interquartile range. Categorical variables were presented as absolute numbers and relative frequencies expressed as percentages.

The normality of the variables was verified using the Shapiro–Wilk test.

The comparison between the expression levels of individual miRNAs and the presence of gadolinium-enhancing lesions at baseline MRI was performed using the Mann–Whitney U test.

To analyse the association between miRNA expression levels and continuous variables (both at baseline and collected longitudinally), a Pearson or Spearman correlation test was initially performed, as appropriate, and—in the case of a statistically significant result—a partial correlation test was performed, controlling for potential confounding factors (age, sex, EDSS, presence of gadolinium-enhancing lesions at baseline, and concomitant disease-modifying drug therapy).

The association between the expression of individual miRNAs and the occurrence of clinical and MRI outcomes of interest (e.g., relapses, confirmed increase in EDSS score and MRI activity) was tested using survival analysis.

To this end, the levels of individual miRNAs were dichotomised into:-High or low expression levels based on median values (for miRNAs expressed in at least 50% of patients)-Present or absent for miRNAs expressed in less than 50% of patients.

Survival curves, stratified according to the expression levels of the candidate miRNA, were estimated using the Kaplan–Meier method and compared using the log-rank test.

For associations that were statistically significant in the univariate analysis, a multivariate Cox model was applied, including covariates such as age, sex, EDSS score, presence of gadolinium-enhancing lesions at baseline, and concomitant therapy with disease-modifying drugs.

A *p*-value < 0.05 was considered significant for a two-tailed test. Statistical analysis was performed using Jamovi software version 2.6.

## 5. Conclusions

This study provides new promising preliminary data from the prospective analysis of the relationship between serum levels of specific miRNAs and disease severity measures in a cohort of patients with a recent MS diagnosis. In particular:miR-34a-5p is associated with the presence of gadolinium-enhancing brain or spinal lesions on MRI and is independently correlated with reduced brain volumes.Increased expression of miR-30b-5p independently correlates with annual brain volume loss.Absent or reduced expression of miR-140-5p seems to increase the risk of MRI activity during follow-up.

These results highlight the importance of circulating miRNAs as potential predictive and prognostic tools in MS management, paving the way for future replication and validation studies in larger cohorts.

## Figures and Tables

**Figure 1 ijms-26-08597-f001:**
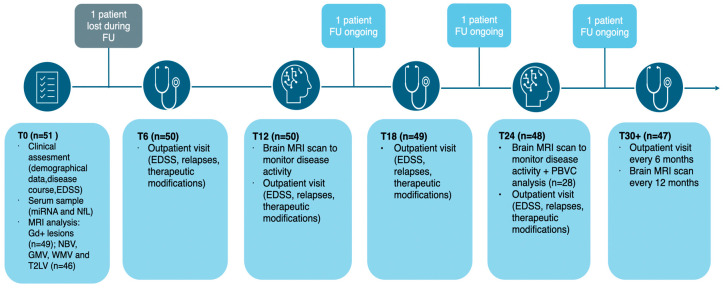
Flowchart illustrating the different phases of the prospective study. Abbreviations: EDSS (expanded disability status scale); FU (follow-up); GMV (gray matter volume); NBV (normalized brain volume); PBVC (percentage brain volume change); T2LV (T2 lesion volume); WMV (white matter volume).

**Figure 2 ijms-26-08597-f002:**
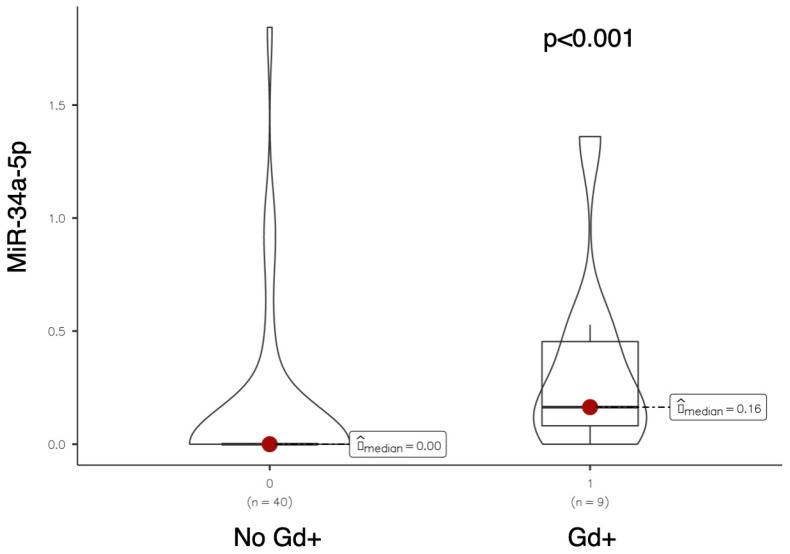
Violin plot comparing miR-34a-5p expression levels between patients without and with gadolinium-enhancing (Gd+) brain MRI lesions at baseline.

**Figure 3 ijms-26-08597-f003:**
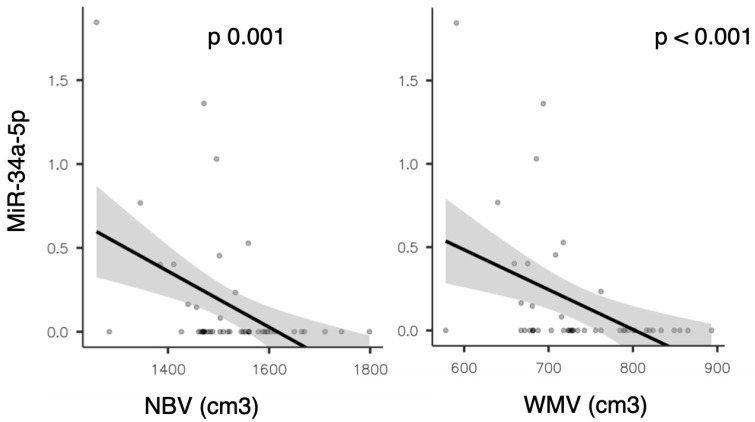
Scatter plots showing the correlation between miR-34a-5p expression and normalized brain volume (NBV) on the (**left**), and white matter volume (WMV) on the (**right**).

**Figure 4 ijms-26-08597-f004:**
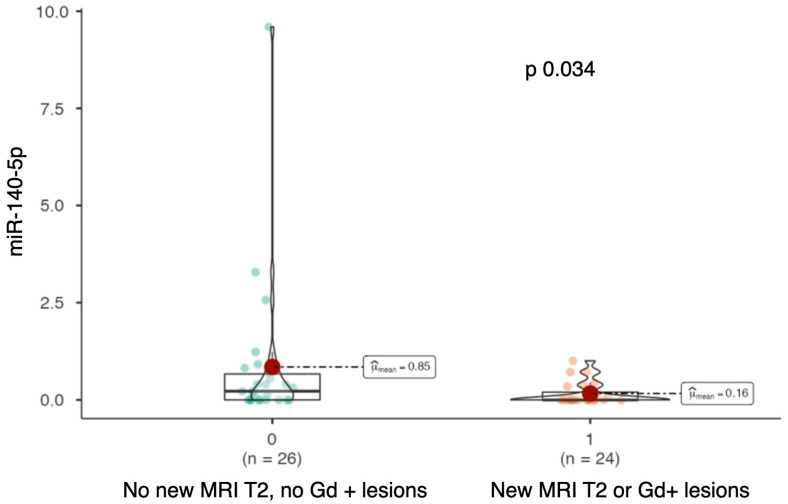
Violin-plot comparing miR-140-5p expression levels between patients without new T2 or gadolinium-enhancing (Gd+) lesions during follow-up and those who developed new lesions.

**Figure 5 ijms-26-08597-f005:**
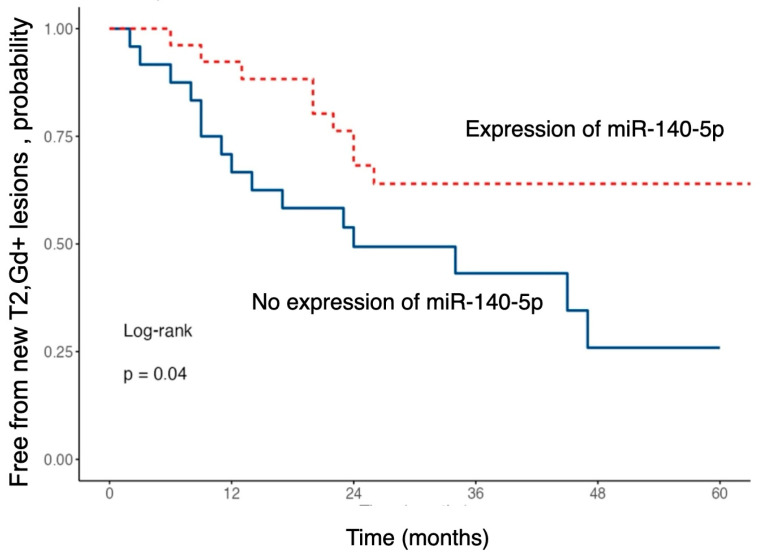
Kaplan–Meier curves showing the probability of remaining free from new T2 or gadolinium-enhancing (Gd+) brain MRI lesions during follow-up, stratified by baseline miR-140-5p expression.

**Figure 6 ijms-26-08597-f006:**
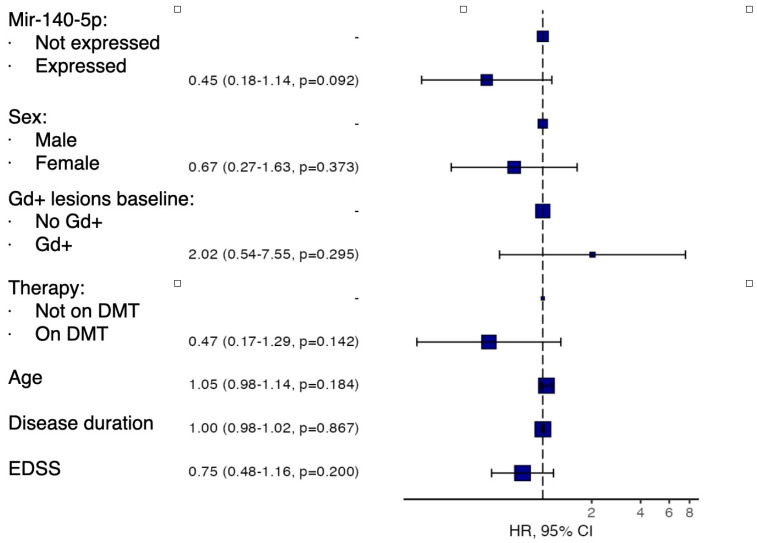
Forest plot showing multivariate analysis of survival without new T2 or gadolinium-enhancing (Gd+) lesions during follow-up according to baseline miR-140-5p expression levels, adjusted for age, sex, baseline Gd+ lesions, disease-modifying therapy, disease duration, and baseline EDSS.

**Figure 7 ijms-26-08597-f007:**
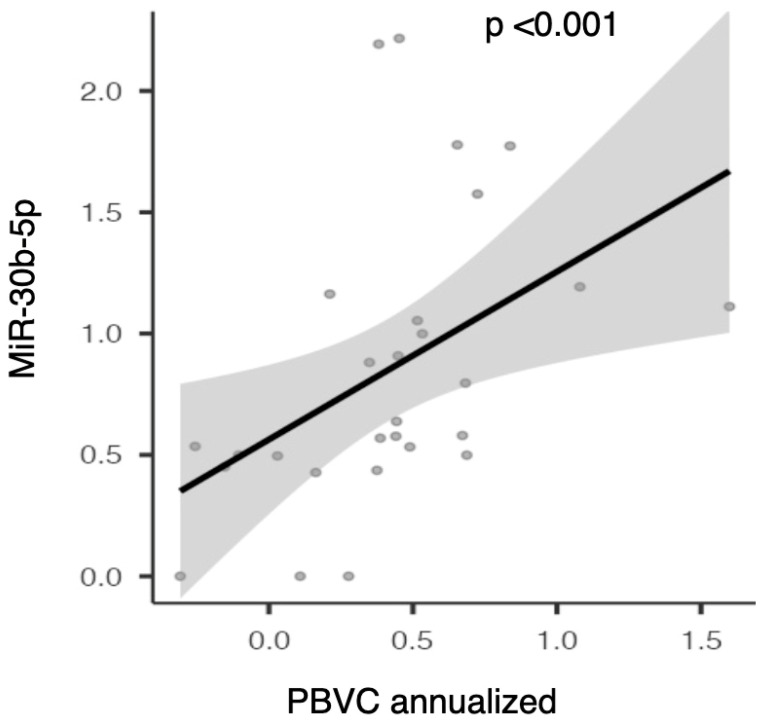
Scatter plot illustrating the correlation between serum miR-30b-5p expression and annualized percentage brain volume change (PBVC).

**Figure 8 ijms-26-08597-f008:**
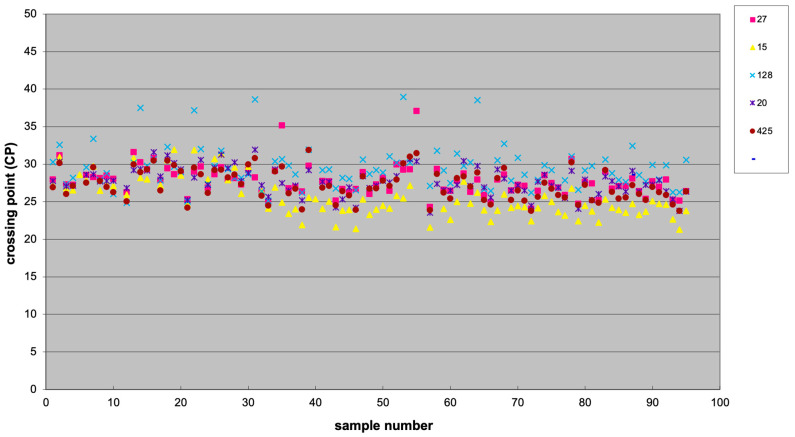
Scatter plot showing the expression of five potential normalizer miRNAs, including miR-27a-3p (27), miR-15b-5p (15), miR-128-3p (128), and miR-20a-5p (20), and miR-425-5p (425).

**Table 1 ijms-26-08597-t001:** Comparison of demographic, clinical, and MRI characteristics between the PBVC subgroup and total study sample. Abbreviations: DMT (disease-modifying treatment); EDSS (expanded disability status scale); FU (follow-up); GMV (gray matter volume); NBV (normalized brain volume); PBVC (percentage brain volume change); PPMS (primary progressive multiple sclerosis); SPMS (secondary progressive multiple sclerosis); T2LV (T2 lesion volume); WMV (white matter volume).

	Total (*n* = 51)	PBVC Cohort (*n* = 28)	*p*
Sex—*n* (%)			
Male	18 (35.3%)	9 (32.1%)	0.603
Female	33 (64.7%)	19 (67.9%)	
Age, years (mean ± SD)	33 ± 7	32 ± 7	0.079
EDSS score (median, range)	1.5(0–4)	2 (0–3.5)	0.122
Disease course—*n* (%)			
RRMS	45 (88.2%)	25 (89.3%)	0.797
PPMS or SPMS	6 (11.8%)	3 (10.7%)	
Disease duration from onset (months), median (range)	10 (1–236)	9 (1–131)	0.583
Baseline MRI with at least one Gd+ lesion *n* (%)	9 (18.4%)	7 (25.9%)	0.226
T2LV cm^3^ (median, range)	2.51 (1.26–1.80)	2.51 (0.21–23.84)	0.578
NBV cm^3^ (median, range)	1523.76 (1258.60–1798.47)	1495.06 (1258.60–1710.68)	0.092
GMV cm^3^ (median, range)	786.07 (667.86–1003.82)	791.17 (667.86–868.05)	0.854
WMV cm^3^ (median, range)	724.89 (578.08–893.080)	716.47 (578.08–855.63)	0.083
Clinical relapse during FU *n* (%)	18 (36%)	11 (39%)	0.642
MRI activity during FU *n* (%)	24 (48%)	13 (46.4%)	0.998
EDSS increase during FU *n* (%)	15 (30%)	11 (39.3%)	0.098
Initiation of DMT during FU *n* (%)	45 (88.2%)	26 (92.9%)	0.087
Median FU duration from study enrolment, months (range)	45 (17–75)	50 (17–65)	0.063

## Data Availability

Access to research data is restricted to authors for privacy reasons; data sharing for motivated reasons can be requested to the corresponding author.

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
