# Peer review of "Serum Levels of miR-34a-5p, miR-30b-5p, and miR-140-5p Are Associated with Disease Activity and Brain Atrophy in Early Multiple Sclerosis"

_ijms, 2025, doi:10.3390/ijms26178597_

Round 1

Reviewer 1 Report

Comments and Suggestions for Authors

In the present research paper Orlando and coworkers analyzed circulatory miRNAs as possible biomarkers in Sclerosis Multiple. In my opinion, the design of the study is quite confusing and not focalized, and for this reason the article does not sound scientifically sufficient for a publication.

In particular:

  1. Usually the exploratory phase is used to screen miRNAs to be validated in the validation phase. In this paper, on the converse, the exploratory phase is used to test some selected miRNAs, and the validation phase to validate one of them and other miRNAs not quantified in the exploratory phase. In my opinion it makes no sense.
  2. How miRNAs to quantify were selected? Both for exploratory and validation study?
  3. There are not any basic information regarding methods of miRNAs extraction (column-based?) and real time PCR (instrument? Amplification cycles?),
  4. In the abstract the patients are defined as MS patients or CIS, whereas in the main text as MS only. Please explain. In particular, in my opinion it is not correct to put together MS and CIS, as CIS can not to develop MS.
  5. Authors wrote for enrolled patients “no current or previous DMT”. What does “previous” mean? In a scientific article, the authors must write precise and specific value.
  6. There is confusion regarding the number of enrolled subjects: in abstract and Methods, patients of exploratory phase are 74, but Table 1 reports 54 patients.
  7. why is miR-140-5p not cited in the title?
  8. Material and Methods section, lines 281-282: the authors wrote that an inclusion criteria for exploratory cohort is “brain MRI scan performed within two years of MS diagnosis”. But in Results, lines 81-82, the authors wrote the 54 [out of 74] patients were identified for whom a brain MRI was available. What is it possible?
  9. The formatting of Table 2 is not acceptable.
  10. Table 1: the range of disease duration for progressive patients is 0-284 months. How is it possible, if the authors wrote as inclusion criteria: “diagnosis within the previous two years”?
  11. In results authors wrote that mean age of exploratory phase patients is 37 years, whereas in table 1 is reported as 38 years. In general, please carefully check the entire paper.

Actually, based on these big and important lacking on study design and subjects enrollment, I prefer not to analyze the results and discussion, because in my opinion the article is based on an incorrect or otherwise not-right structured premise.

Author Response

We thank this reviewer for the opportunity of clarifying some aspects of the manuscript.

Reviewer 2 Report

Comments and Suggestions for Authors

Reviewer Report

 Summary of the manuscript:

This study is scientifically justified, as it explored the expression of 18 miRNAs as disease and progression in multiple sclerosis (MS) in an exploratory and validation cohort. Its focus on trying to identify alternative RNA-based methods to track disease and progression addressed a significant gap in current knowledge. The study found that miR-34a-5p, miR-140-5p, and miR-30b-5p were markers of disease activity and progression in patients with a recent diagnosis of MS and could have both diagnostic and therapeutic implications for this group of patients.

General comments:

Whilst the manuscript addresses an important topic in the diagnosis and prognosis of MS and the findings, while not novel, add on to the growing pool of evidence for RNA-based biomarkers for MS. However, the writing could be clearer throughout the manuscript and significant improvements need to be made. For example: in the introduction section, no reference was made to studies that have investigated serum concentration of circulating miRNAs, allowing the identification of different alteration profiles in MS patients. Whilst the authors state that they chose their candidate miRNA targets from literature, they did not cite the studies.  Furthermore, the methodology section is quite lacking in critical details. The participants in either cohort were diagnosed using different criteria; there is no evidence of the suitability of the qPCR normaliser used; it is unclear who comprised the control groups in both cohorts; and a two-year window for a brain scan (as part of inclusion criteria) may be problematic as it may miss changes that happened closer to the time when the symptoms started. The results section should also be significantly improved, as meaningful disease v control comparisons have not been included and/or properly presented in a manner that facilitates easy reading for the reader.

Introduction section

The paper would greatly benefit from the authors discussing the current methods used to measure the pathological processes active in individual patients, with the goal of predicting the clinical evolution of the disease. Additionally, the authors need to highlight the shortcomings of these current methods, and why there is a need for miRNA-based approaches and how these would mitigate the shortcomings of currently adopted methods. There isn’t enough reference to literature in the introduction section to clearly articulate the rationale for this study.

Methods

Study design

The idea for having an exploratory and validation cohort is to basically confirm what you see in one cohort in the other cohort. As such, it is important that individuals in both these cohorts are “similar enough” for this comparison to be done without introducing variability that may bias the results. It would benefit the reader if the authors included a table describing both cohorts, so variables can be compared across these cohorts.  

Diagnostic criteria

MS Diagnosis differed for both cohorts (2010 vs 2017 McDonald criteria). This makes it challenging to directly compare findings from the two cohorts, as the patient classification system is not the same. If possible, the authors could retrospectively recategorize their exploration cohort using the  2017 McDonald criteria, then re-analyse their  data. If not possible, the authors need to clearly detail how this may have impacted their findings, by highlighting the keys differences in the two diagnostic criteria.

Suggested analysis

MS is an immune-mediated disease, and as such, this study would have greatly benefited from the inclusion of cytokine profiling. Although other clinical and radiological markers of disease were part of the study, cytokines may have shed more light on the immune mechanisms driving the disease process. For example, cytokine profiling would have enabled the authors to see if changes in miRNA expression levels reflected alterations in cytokine expression levels (immune activation), especially in MS where inflammation plays a central role in the disease. If cytokine profiling is still possible (even for a select number of participants), it would enhance the scientific and clinical impact of your study and authors should consider doing it. If not feasible, it should at least be mentioned as a limitation, with a suggestion to explore in future work.

Results

For the exploratory phase, the authors gave a clear breakdown of the groups to which participants belonged. However, this was not done for the validation phase. It is important that the same subgroups in the exploratory study are present in the validation phase for direct comparison of findings. For example, of the 51 patients in the validation phase, a clear breakdown of their disease group should be given, just like in the exploratory study (relapsing-remitting disease course OR progressive disease course)

Minor concerns:

  • Line 60 - No reference at all has been made to studies that have investigated serum concentration of circulating miRNA, allowing the identification of different alteration profiles in MS patients.
  • Line 97-98 – this should be one sentence.
  • Line 180 – Annualized PBVC should be clearly defined. Does it represent only a LOSS in brain function, because that’s what Figure 7, and the x-axis caption suggest. Whilst it’s rare that PBVC goes up, there are circumstances that it does, especially in individuals who have started therapy, where the brain shrinks, then increases again in volume slightly.
  • Line 281 - A two-year window for a brain scan may miss changes that happened closer to the time when the symptoms started, hence the accuracy of the biomarker may be affected. The authors should discuss this as a possible limitation.
  • Line 283 - authors mention disease-modifying therapies (DMTs) but don’t give examples.
  • Line 286 -  reference should be made to the literature that formed the basis for selection of the 10 miRNAs targeted in the exploration study. This should also be done for the eight different miRNAs targeted in the validation phase.
  • 18 miRNAs in total were targeted as biomarkers of MS. What were the findings regarding the expression of those miRNAs in the control groups? That was not mentioned for the exploration phase.
  • Line 288 - The control group of non-MS neurological disorders includes what kind of disorders? Did it also include people with no neurological disorders? This is not clear from the manuscript. The authors should also include a table of characteristics comparing variables between their patient and control groups.
  • miR-425-5p was determined to be an appropriate normaliser using BestKeeper software. What parameters were used to determine appropriateness? Were you looking a correlation, CV, SD? A diagram showing no significant difference in the mean miR-425-5p Ct values across comparison groups should be included, to confirm the validity of qPCR and the study findings.
  • The data analysis section is not very clear. Please describe in detail what statistical tests were done and how comparisons were made.
  • Table 2 has numbers from 1-60 which are very confusing as to what they are. Additionally, the variables being compared between the study participant groups should also have column in the table for p-values.
  • If miRNA expression comparisons were done between the MS and control groups, it might be worth having images to show the findings, possibly as boxplots. Additionally, the y-axis on Figures 2 and 3 just have the miRNA expression there, but the manuscript doesn’t say how that expression was calculated. Did they use the Livak and Schmittgen method to calculate expression? Are they reporting fold differences in expression? This needs to be clear in the manuscript. Basically, when the authors say this miRNA’s expression is higher in a certain group, what calculations did they do to determine higher or lower expression of the miRNAs? In Line 354–355, authors wrote - “miRNA levels were expressed as the ratio of candidate miRNA concentration to the normalizer.” Which method is this and please cite the source?
  • Line 360 – The study actually analysed 18 miRNAs, and not 10. There are no references to the studies that brought about the choice of miRNA targets for study. Of the 18 candidates, only 4 gave results. Did the other 14 not amplify or findings were not statistically significant?

Author Response

(The authors gave the same response as above.)

Round 2

Reviewer 1 Report

Comments and Suggestions for Authors

In my opinion it is scientifically wrong to focalize on different miRNAs in retrospective and prospective studies of a single paper.

Author Response

The reason for including both the retrospective and prospective studies in the same paper is to emphasize the better chance of the prospective design to detect significant associations between biomarker expression and clinical data compared to the retrospective design.

However, if this reviewer feels it necessary for publication we are open to remove the retrospective part of the manuscript.

Reviewer 2 Report

Comments and Suggestions for Authors

Thank you for thoroughly addressing the majority of the concerns raised in the previous round of reviews. Just a few minor things below to pay attention to:

1. Line 67-70 need to be rephrased for clarity.

2. Revise Line 93-95, there is a missing word.

3. Replace the word “subjects’ with “participants”. We cannot use subjects to describe individuals that took part in our studies.

4. The authors mentioned but did not include Figure 8 in their manuscript as stated. As such, I cannot comment on the appropriateness of the qPCR data normaliser used.

5. Whenever a commercial reagent and/or instrument is mentioned in the manuscript, it should be accompanied by the manufacturer and geographical location in brackets.

6. Please indicate on which instrument the qPCR was performed?

7. Please provide a reference or article source for the 2−ΔCt method.

8. If the significant p-values in the tables are going to be bold , make sure this is the case for all significant p-values. See Table 1.

Author Response

1. Line 67-70 need to be rephrased for clarity.

The sentence has been rephrased as follows: "However, most of these variables are non-scalable and retrospective risk factors, as for demographic features and clinical presentation, which are not particularly helpful as predictive factors at the individual level. Other available measures are not linked to specific biological processes or not frequently repeatable over time due to acceptability, time and cost constraints, as in the case of MRI, conventional cerebrospinal fluid analysis, and evoked potentials".

2. Revise Line 93-95, there is a missing word.

Thank you for this suggestion. The entire sentence has been rephrased for clarity as follows: "Evidence on the utility of miRNAs expression to assess treatment efficacy is scarce and controversial. Karimi et al. showed similar levels of miR-326-3p in blood lymphocytes of MS patients with and without good clinical response to interferon-beta therapy, in contrast to previous reports showing miR-326-3p down-regulation in peripheral blood mononuclear cells of interferon-beta responders"

3. Replace the word “subjects’ with “participants”. We cannot use subjects to describe individuals that took part in our studies.

"Subjects" has been replaced throughout the manuscript 

4. The authors mentioned but did not include Figure 8 in their manuscript as stated. As such, I cannot comment on the appropriateness of the qPCR data normaliser used.

Figure 8 has been correctly inserted in the manuscript

5. Whenever a commercial reagent and/or instrument is mentioned in the manuscript, it should be accompanied by the manufacturer and geographical location in brackets.

These details have been added to the manuscript.

6. Please indicate on which instrument the qPCR was performed?

Real-time PCR was performed on the CFX Connect Real-Time System (Bio-Rad, Hercules, CA, USA). This information has been added to the Methods section.

7. Please provide a reference or article source for the 2−ΔCt method.

The following reference has been added: Livak KJ, Schmittgen TD. Analysis of relative gene expression data using real-time quantitative PCR and the 2(-Delta Delta C(T)) Method. Methods. 2001 Dec;25(4):402-8. doi: 10.1006/meth.2001.1262.

8. If the significant p-values in the tables are going to be bold, make sure this is the case for all significant p-values. See Table 1.

All statistically significant p-values (i.e. <0.05) are now shown in bold throughout the tables (supplementary table 1 report no significant p values).

Round 3

Reviewer 1 Report

Comments and Suggestions for Authors

Yes, to me it is necessery to remove the retrospective part of the manuscript.

Author Response

Agreed

Round 4

Reviewer 1 Report

Comments and Suggestions for Authors

The authors answered my queries.